# *KRAS^G12C^* Inhibitor as a Treatment Option for Non-Small-Cell Lung Cancer with Comorbid Interstitial Pneumonia

**DOI:** 10.3390/cancers16071327

**Published:** 2024-03-28

**Authors:** Kazushi Fujimoto, Satoshi Ikeda, Erina Tabata, Taichi Kaneko, Shinobu Sagawa, Chieri Yamada, Kosumi Kumagai, Takashi Fukushima, Sanshiro Haga, Masayuki Watanabe, Tatsuya Muraoka, Akimasa Sekine, Tomohisa Baba, Takashi Ogura

**Affiliations:** Department of Respiratory Medicine, Kanagawa Cardiovascular and Respiratory Center, 6-16-1 Tomioka-higashi, Kanazawa-ku, Yokohoma 236-0051, Japan; fujimoto.8x50n@kanagawa-pho.jp (K.F.); tabata.19061@kanagawa-pho.jp (E.T.); kaneko.9150n@kanagawa-pho.jp (T.K.); sagawa.8z30n@kanagawa-pho.jp (S.S.); yamada.9040n@kanagawa-pho.jp (C.Y.); kumagai.8y20n@kanagawa-pho.jp (K.K.); fukushima.9u20m@kanagawa-pho.jp (T.F.); haga.8w40n@kanagawa-pho.jp (S.H.); watanabe.8v30n@kanagawa-pho.jp (M.W.); muraoka.1650m@kanagawa-pho.jp (T.M.); sekine.1v50c@kanagawa-pho.jp (A.S.); baba.19049@kanagawa-pho.jp (T.B.); ogura.0302b@kanagawa-pho.jp (T.O.)

**Keywords:** non-small-cell lung cancer, interstitial pneumonia, idiopathic pulmonary fibrosis, acute exacerbation, pneumonitis, cytotoxic drug, immune checkpoint inhibitor, *KRAS^G12C^*, radiation therapy

## Abstract

**Simple Summary:**

Interstitial pneumonia (IP) represents a significant prognostic comorbidity in patients diagnosed with non-small-cell lung cancer (NSCLC). Even if treatment for lung cancer proves effective, the occurrence of an acute exacerbation in pre-existing IP can pose a substantial risk to patient survival. Consequently, it is crucial to opt for a therapy that minimizes the likelihood of triggering an acute IP exacerbation. Evidence is limited regarding pharmacotherapy for advanced NSCLC with a low risk of triggering the exacerbation of pre-existing IP, but *KRAS^G12C^* inhibitors may be a potential treatment option for NSCLC with comorbid IP. In this review article, we discuss the promise and prospects of molecular-targeted therapy, particularly *KRAS^G12C^* inhibitors, which gained little attention as a treatment for NSCLC with comorbid IP.

**Abstract:**

Non-small-cell lung cancer (NSCLC) with comorbid interstitial pneumonia (IP) is a population with limited treatment options and a poor prognosis. Patients with comorbid IP are at high risk of developing fatal drug-induced pneumonitis, and data on the safety and efficacy of molecularly targeted therapies are lacking. *KRAS* mutations have been frequently detected in patients with NSCLC with comorbid IP. However, the low detection rate of common driver gene mutations, such as epidermal growth factor receptor and anaplastic lymphoma kinase, in patients with comorbid IP frequently results in inadequate screening for driver mutations, and *KRAS* mutations may be overlooked. Recently, sotorasib and adagrasib were approved as treatment options for advanced NSCLC with *KRAS^G12C^* mutations. Although patients with comorbid IP were not excluded from clinical trials of these *KRAS^G12C^* inhibitors, the incidence of drug-induced pneumonitis was low. Therefore, *KRAS^G12C^* inhibitors may be a safe and effective treatment option for NSCLC with comorbid IP. This review article discusses the promise and prospects of molecular-targeted therapies, especially *KRAS^G12C^* inhibitors, for NSCLC with comorbid IP, along with our own clinical experience.

## 1. Introduction

Non-small-cell lung cancer (NSCLC) with comorbid interstitial pneumonia (IP) presents limited treatment options and a poor prognosis due to the difficulties of treating NSCLC itself and the increased risk of acute exacerbation of IP with NSCLC treatment. Reportedly, 5–10% of NSCLC patients have IP as a comorbidity at the time of diagnosis of NSCLC [1,2,3]. Conversely, the prevalence of lung cancer in idiopathic pulmonary fibrosis (IPF) ranges from 2.7% to 48% [4], which is significantly higher than that in the general population [5]. This is due to the fact that the presence of IP, especially pulmonary fibrosis, has been shown to be associated with lung carcinogenesis [6]. The cumulative incidence rates of lung cancer in patients with IPF were 3.3%, 15.4%, and 54.7% at 1, 5, and 10 years, respectively. Additionally, the coexistence of emphysema may increase the risk of lung cancer development in patients with IPF [7,8]. Unlike the general population, squamous cell carcinoma is the most frequent type of lung cancer in patients with IPF [4,7]. Lung cancer is a poor prognostic factor in IPF, with 11% of all deaths attributed to it in Japan [9]. Furthermore, Kato et al. revealed that the 1-, 3-, and 5-year all-cause mortality rates in patients with IPF after a lung cancer diagnosis were 53.5%, 78.6%, and 92.9%, respectively [7]. Both IPF itself and the development of lung cancer demonstrated a significant effect on prognosis.

Acute exacerbation is an abrupt worsening of IP, sometimes clearly triggered by factors, such as infection or surgery, leading to high short-term mortality [10]. The criteria for diagnosing acute exacerbations in IPF proposed by an international working group report [11] are as follows: (1) previous or concurrent diagnosis of IPF; (2) acute worsening or development of dyspnea, typically within a 1-month duration; (3) computed tomography (CT) findings revealing new bilateral ground-glass opacity and/or consolidation superimposed on a background reticular shadow or honeycomb lung; and (4) deterioration that is not fully explained by cardiac failure or fluid overload. The annual incidence of acute exacerbations in IPF ranges from 8.5% to 14.2% [12,13,14], preceding up to 46% of deaths in IPF [9,11]. The median survival post-exacerbation is approximately 3–4 months, with high in-hospital mortality frequently exceeding 50% [11]. Acute exacerbations in IP other than IPF have a poor prognosis but are still better than IPF. Miyashita et al. [15] revealed that the 90-day mortality after acute exacerbation was 57% in IPF, 29% in non-IPF idiopathic IPs, and 33% in secondary IP. Patients with IP other than IPF demonstrated a significantly better survival rate than those with IPF after acute exacerbations (*p* < 0.001). Among those who survived, the proportion of patients needing long-term oxygen therapy post-exacerbation was 63% for IPF, 35% for non-IPF idiopathic IPs, and 46% for secondary IP.

Cytotoxic chemotherapy has the potential to trigger acute exacerbations of IP. The incidence of acute exacerbation was as high as 30% in cytotoxic chemotherapy in cases with the UIP pattern, and nearly all of these cases were graded as severe (grade 3 or higher) [16]. A nationwide surveillance study in Japan revealed that acute exacerbations of idiopathic IP related to cytotoxic chemotherapy for lung cancer occurred in 13.1–22% of the overall population [16,17]. As for first-line treatment for NSCLC with comorbid IP, platinum combination therapies such as carboplatin plus nanoparticle albumin-bound paclitaxel (nab-paclitaxel) are frequently used as the standard therapy, according to results from several prospective intervention studies [18,19,20]. However, regarding second-line and subsequent treatment for NSCLC with comorbid IP, therapy with universal consensus as the standard of care remains unavailable due to insufficient evidence.

Recently, immune checkpoint inhibitors (ICI) have demonstrated better efficacy for some NSCLC cases and are considered key drugs for improving the survival of NSCLC patients [21,22,23]. However, drug-induced pneumonitis is recognized as one of the immune-related adverse events (irAE) and can be life-threatening, even in patients without pre-existing IP. Nevertheless, the incidence of acute exacerbations or drug-induced pneumonitis related to ICI in NSCLC patients with comorbid IP has not been firmly established because patients with IP are often excluded from large-scale prospective studies. There have been a few reports regarding the efficacy and safety of ICI for NSCLC with comorbid IP, but the data have varied between reports. For nivolumab, the incidence of ICI-induced pneumonitis in NSCLC patients with comorbid IP was lower, with zero cases reported in the pilot study [24] and 11.1% in the phase II study [25]. However, the efficacy was higher compared with the non-IP group [26]. Conversely, one phase II study for atezolizumab in previously treated NSCLC patients with comorbid IP was discontinued because of the high incidence of ICI-induced pneumonitis. The incidence rates were 29% for all grades, 24% for grades ≥ 3, and 6% for grade 5 [27]. As for ICI, there are relatively few reports regarding its safety for NSCLC with comorbid IP, as described above, and it has not become the standard of care for NSCLC with comorbid IP because of concerns about irAE pneumonitis.

Radiotherapy for NSCLC patients with comorbid IP should be carefully considered because of the high risk of triggering acute exacerbations. Patients with visible fibrosis on chest X-rays demonstrated a significantly higher rate of postradiation death than those without fibrosis (relative risk 165.7) [28]. Given the poor prognosis after radiation therapy, chemotherapy remains the primary treatment option for unresectable and locally advanced NSCLC patients with comorbid IP.

Interstitial lung abnormality (ILA) has been reported to be a factor affecting the prognosis of NSCLC and the risk of adverse events during NSCLC treatment. This includes even very minor interstitial shadows that are incidentally detected on CT scans conducted for screening purposes related to other lung and extrapulmonary abnormalities. ILA is defined as non-dependent abnormalities affecting >5% of any lung zone, as per the position paper of the Fleischner Society [29]. ILA includes various features such as ground-glass opacities, reticular abnormalities, lung distortion, traction bronchiectasis, honeycombing, and nonemphysematous cysts. A population-based cohort study in the United States revealed that high attenuation areas (HAA), which encompass some of the ILA features listed above, increased in prevalence with higher categories of pack years. HAA increased by 2.5 cm^3^ (95% confidence interval [CI]: 1.8–3.3 cm^3^) for every 10 cigarette pack years after adjusting for age, sex, and ethnicity [30]. The prevalence of ILA detected by CT has ranged from 8% to 22.8% in cohorts of smokers [31,32], and 13.2% of patients had ILA at the time of NSCLC diagnosis [33]. In a study involving patients with stage IV NSCLC, a high ILA score, which was visually evaluated, was associated with shorter survival [34]. Nakanishi et al. examined pretreatment chest CT scans for ILAs in 83 patients treated with anti-programmed death-1 (PD-1) antibodies such as nivolumab or pembrolizumab. They revealed a high incidence of immunotherapy-associated pneumonitis at 17% (*n* = 14), with pre-existing ILAs correlating with a six-fold increase in the risk of drug-associated pneumonitis, especially those displaying a predominant ground-glass pattern of pneumonitis [35].

The challenge is selecting chemotherapy regimens with a lower risk of triggering acute exacerbations. However, only a limited number of chemotherapy regimens have been evaluated for their low risk of causing acute exacerbations. As patients with IP have been excluded from the majority of prospective clinical studies, there is very limited evidence regarding pharmacotherapy for NSCLC patients with comorbid IP.

## 2. Driver Gene Mutations in NSCLC with Comorbid IP

First-line therapy with tyrosine kinase inhibitors (TKI) targeting the respective gene mutations is usually recommended for NSCLC with driver gene mutations such as the epidermal growth factor receptor (*EGFR*), anaplastic lymphoma kinase (*ALK*), *BRAF*, and *ROS1* genes. However, there are several patients with advanced or recurrent NSCLC whose driver genes have not been sufficiently identified before initial pharmacotherapy, especially in patients with comorbidities such as IP. A retrospective real-world study conducted in Japan, which aimed to investigate the actual status of biomarker testing and drug therapy, revealed that multiplex gene testing was conducted in 47.7% of patients with advanced or recurrent NSCLC. Among these, next-generation sequencing (the Oncomine DX Target Test) was conducted in 67.2% of multiplex gene testing. The rates of biomarker testing implementation for specific genetic alterations were as follows: *EGFR* at 84.2%, *ALK* at 78.8%, *ROS1* at 72.8%, *BRAF* at 54.3%, and *MET* at 54.4%. Furthermore, in adenocarcinoma cases, the prevalence of each driver mutation, including *EGFR*, *ALK*, *ROS1*, *BRAF*, and *MET*, was found to be 34.0%, 3.2%, 2.1%, 1.2%, and 1.6%, respectively. Additionally, in non-adenocarcinoma cases, the prevalence was found to be 3.7%, 1.6%, 0.3%, 0.8%, and 2.1%, respectively. Factors associated with the lack of multiplex gene testing were Eastern Cooperative Oncology Group Performance Status of 3 or 4 (odds ratio (OR): 0.47; 95% CI: 0.32–0.70; *p* < 0.005), comorbidities (OR: 0.54; 95% CI: 0.44–0.67; *p* < 0.005), and non-adenocarcinoma (OR: 0.70; 95% CI: 0.56–0.87; *p* < 0.005) in the multivariate model. Among patients with comorbidities, 10.5% had interstitial lung disease [36].

Pre-existing IP is considered one of the risk factors for drug-induced pneumonitis caused by molecular-targeted therapy, especially by *EGFR*-TKIs, along with factors such as older age, smoking history, and poor performance status. Japanese nationwide surveillance of NSCLC with comorbid IP by Minegishi et al. revealed that the incidence rates of acute exacerbation after *EGFR*-TKI were 83.3% for first-line pharmacotherapy and 44.4% for second-line pharmacotherapy [17]. Gefitinib was known to have a higher incidence rate of drug-induced pneumonitis than conventional chemotherapy [37]. The observed naive cumulative incidence rates at the end of a 12-week follow-up were 4.0% (95% CI: 3.0–5.1%) for gefitinib and 2.1% (95% CI: 1.5–2.9%) for chemotherapy. The overall OR for gefitinib versus chemotherapy, adjusted for imbalances in risk factors between treatments, was 3.2 (95% CI: 1.9–5.4). Similarly, other *EGFR*-TKIs, such as erlotinib [38] and osimertinib [39], exhibited a higher frequency of drug-induced pneumonitis during molecular-targeted therapies in NSCLC with comorbid IP compared with those without IP. A prospective cohort study of drug-induced pneumonitis after erlotinib administration in patients with *EGFR* mutation-positive NSCLC identified pre-existing IP and low residual normal lung function (50% or less) as risk factors for developing drug-induced pneumonitis. Specifically, low residual normal lung function (≤50%) was recognized as a risk factor for developing fatal drug-induced pneumonitis [38]. Gemma et al. reported the results of real-world data about osimertinib for *EGFR T790M*-positive NSCLC and revealed IP history or coexistence as factors that are potentially associated with the onset of drug-induced pneumonitis during the treatment (adjusted OR: 3.51; 95% CI: 2.10–5.87) [39].

Pulmonary toxicity in patients taking *ALK*-TKIs has been reported to be relatively rare, with incidence rates of 1.8%, 1.1%, and 2.6% for crizotinib, ceritinib, and alectinib, respectively, regardless of the coexistence of IP [40]. However, data to fully report the efficacy and incidence of drug-induced pneumonitis for molecular-targeted therapies targeting driver gene mutations other than *EGFR* in NSCLC patients with comorbid IP remain insufficient. Therefore, further study accumulation is required.

The prevalence of driver gene mutations related to molecular-targeted therapy is reported to be low in the NSCLC population with comorbid IP. A retrospective study conducted in a single center in Japan [41] revealed that only 0.4% (1/246) of patients with *EGFR* mutations demonstrated pre-existing IP, and 3.2% (1/31) of patients with pre-existing IP exhibited *EGFR* mutations. The rate of *EGFR* mutation in patients with pre-existing IP was lower than that in patients without IP (1/246 vs. 30/309) [41]. The prevalences of driver gene mutations in NSCLC patients with comorbid IP classified as UIP patterns were 1.9%, 20.4%, and 3.7% for *EGFR*, *KRAS*, and *BRAF*, respectively [42]. *EGFR* and *KRAS* mutations were observed in 9 (14.1%) and 24 (37.5%) cases, respectively, in lung adenocarcinoma with smoking-related IP [43]. Notably, *BRAF* mutations demonstrated a significantly higher frequency in patients with lung cancer with IPF, accounting for 6 of 35 cases (17.1%) [44], which is much higher than the known prevalence of 2–4% in the general lung cancer population [45,46]. As previously mentioned, a retrospective real-world study conducted in Japan [36] revealed that comorbidities, including IP, were associated with the absence of multiplex gene testing. Consequently, NSCLC patients with comorbid IP were not thoroughly investigated for multiple driver gene mutations, considering the underlying risk of drug-induced pneumonia. However, *KRAS* and *BRAF* were sometimes detected as driver gene mutations even in NSCLC patients with comorbid IP, and these driver gene mutations and treatment opportunities were missed in patients with comorbid IP. Therefore, all NSCLC patients, regardless of comorbidity, should be comprehensively investigated for driver gene mutations.

*KRAS* mutations occupied a significant portion of driver gene mutations in NSCLC with comorbid IP, particularly when presenting a UIP pattern in chest CT. However, the low detection rate of common driver gene mutations, such as *EGFR* and *ALK*, in patients with comorbid IP frequently results in inadequate screening for driver mutations, and *KRAS* mutations may be unintentionally overlooked.

## 3. NSCLC with *KRAS* Mutations

### 3.1. Epidemiology of KRAS Mutations and the Development of KRAS^G12C^ Inhibitors

The prevalence of *KRAS* mutations in NSCLC is distinct from that of other genetic variants. *KRAS* is one of the most prevalent oncogenic drivers in NSCLC, found in over 30% of lung adenocarcinomas based on data from the Cancer Genome Atlas [47]. However, the prevalence varies based on ethnicity and tumor stage and is associated with smoking and female patients [48]. *KRAS* mutations encompass a range of distinct substitutions, with codon 12 being the most frequent site. The *KRAS^G12C^* substitution, occurring in 10–30% of lung adenocarcinoma cases and accounting for 40–50% of all *KRAS* mutations, is prevalent [49,50,51]. In lung squamous cell carcinoma, it is approximately 4.5% [52]. Other notable point mutations include glycine-to-aspartic acid (*KRAS^G12D^*) and glycine-to-valine (*KRAS^G12V^*) substitutions, which are observed in approximately 5% and 4% of patients with lung adenocarcinoma, respectively [49]. Epidemiologically, *KRAS^G12C^* and *KRAS^G12V^* are linked to a history of smoking, whereas *KRAS^G12D^* is associated with nonsmokers [53]. These mutations lead to distinct alterations in downstream signaling pathways, affecting clinical outcomes and treatment responses.

*KRAS* mutations were difficult to target for a long time because the surface structure of the *KRAS* protein was relatively smooth, with limited pockets available for small molecules to bind, apart from the GTP/GDP binding site. The development of GTP-competitive inhibitors is difficult compared with other drug discovery approaches for RAS inhibitors because the strong affinity between *KRAS* and GTP poses challenges for competitive inhibition [54]. However, in 2013, Shokat et al. made a groundbreaking discovery by identifying a switch II pocket in the *KRAS^G12C^* mutation that was amenable to binding with small-molecule compounds [55]. Subsequent discoveries of compounds that irreversibly bind to this pocket and cause allosteric inhibition have resulted in the development of *KRAS^G12C^* inhibitors [56]. Sotorasib and adagrasib (Figure 1) were approved as second-line treatment options for advanced or recurrent NSCLC with *KRAS^G12C^* mutations.

### 3.2. Sotorasib

The phase II portion of the CodeBreaK100 trial [57], which was part of an international phase I/II trial [57,58], administered oral sotorasib at a daily dose of 960 mg to patients with *KRAS^G12C^* mutation-positive NSCLC after undergoing standard therapy. The primary endpoint of the study was the assessment of objective responses as determined by an independent central review. A total of 126 patients were enrolled in the study, with 124 of them having measurable disease according to Response Evaluation Criteria in Solid Tumors (RECIST) version 1.1. The overall response rate (ORR) was 37.1% (95%CI: 28.6−46.2), with 4 patients achieving a complete response and 42 patients demonstrating a partial response (PR). The median duration of response (DOR), disease control rate, progression-free survival (PFS), and overall survival (OS), which were evaluated as secondary endpoints, were 11.1 months (95% CI: 6.9−not evaluable), 80.6% (95% CI: 72.6−87.2), 6.8 months (95% CI: 5.1−8.2), and 12.5 months (95% CI: 10.0−not evaluable), respectively.

Treatment-related adverse events occurred in 88 (69.8%) of 126 patients, with 26 (20.6%) patients experiencing grade ≥ 3 adverse events. No adverse events in grade 5 were observed. An exploratory analysis observed treatment responses in subgroups based on factors such as programmed death ligand-1(PD-L1) expression, tumor mutation burden (TMB), and the comutation status of *STK11*, *KEAP1*, and *TP53*. The results indicated that the PD-L1-negative, TMB-low, *STK11*-mutated, and *KEAP1*-unmutated groups demonstrated greater tumor reduction than the other groups.

The 2-year follow-up data on sotorasib has already been reported [59]. It revealed an ORR of 41%, a median DOR of 12.3 months, a PFS of 6.3 months, an OS of 12.5 months, and a 2-year OS rate of 33%. Long-term clinical benefit, defined as PFS of 12 months or over, was observed in 40 (23%) patients, irrespective of PD-L1 expression levels. Additionally, a subset of patients with somatic *STK11* and/or *KEAP1* alterations, along with lower baseline circulating tumor DNA, also experienced this clinical benefit. Based on these findings, sotorasib was evaluated as well tolerated in the second-line regimen for NSCLC with *KRAS^G12C^* mutations, with minimal late-onset treatment-related adverse events that did not necessitate treatment discontinuation.

In the phase III trial of CodeBreaK200 [60], patients with advanced NSCLC bearing *KRAS^G12C^* mutations who had experienced disease progression after previous platinum-based chemotherapy and treatment with a PD-1 or PD-L1 inhibitor were randomly assigned to two groups. One group received oral sotorasib at 960 mg daily, whereas the other group received docetaxel intravenously (75 mg/m^2^) once every three weeks. Randomization was stratified based on the number of previous lines of therapy for advanced disease, Asian ethnicity, and a history of central nervous system (CNS) metastases. The primary endpoint of the study was PFS, evaluated through a blinded independent central review within the intention-to-treat population. The study successfully met its primary endpoint after a median follow-up of 17.7 months (interquartile range (IQR): 16.4–20.1), demonstrating a statistically significant improvement in PFS for sotorasib compared with docetaxel. Specifically, the PFS was 5.6 months (95% CI: 4.3–7.8) for sotorasib compared with 4.5 months (IQR: 3.0–5.7) for docetaxel. The hazard ratio was 0.66 (IQR: 0.51–0.86), and the *p*-value was 0.0017. Sotorasib was well tolerated, with fewer occurrences of grade ≥ 3 severe adverse events (33% in the sotorasib group vs. 40% in the docetaxel group). Furthermore, fewer serious treatment-related adverse events were observed with sotorasib (11%) than with docetaxel (23%).

### 3.3. Adagrasib

A phase II cohort of the KRYSTAL-1 study [61] evaluated the efficacy of adagrasib at 600 mg orally twice daily in patients with *KRAS^G12C^*-mutant NSCLC who had previously been treated with platinum-based agents and ICIs. The primary endpoint was ORR, as determined by an independent central review, and secondary endpoints included DOR, PFS, OS, and safety. A total of 116 NSCLC patients with *KRAS^G12C^* mutations were enrolled, with 98.3% of them having received prior treatment involving both chemotherapy and immunotherapy. At the time of enrollment, 112 patients had measurable lesions according to RECIST version 1.1, and the ORR was evaluable in 48 (42.9%) of them. The median DOR was 8.5 months (95% CI: 6.2–13.8), and the median PFS was 6.5 months (95% CI: 4.7−8.4). At a median follow-up of 15.6 months, the median OS was 12.6 months (95% CI: 9.2−19.2). Among the 33 patients with previously treated or asymptomatic stable CNS metastases, the confirmed intracranial ORR was 33.3% (95% CI: 18.0−51.8). Treatment-related adverse events were reported in 97.4% of the patients, with 44.8% experiencing grade ≥ 3 events. Of these, two patients experienced grade 5 adverse events, leading to treatment discontinuation for 6.9% of patients.

## 4. Potential of *KRAS^G12C^* Inhibitors as a Treatment Option for NSCLC with Comorbid IP

### 4.1. Significance of Exploring KRAS Mutations in NSCLC with Comorbid IP

The risk factors of *KRAS* mutation were more frequently revealed in patients with IP than in those without IP, independent of smoking history and elder age, based on the data of DNA extracted from bronchiolar metaplasia in honeycomb lesions [62]. Moreover, the *KRAS* mutation was considered to have a strong relationship with *TP53* mutations because both are known to be smoking-related mutations [63]. Guyard et al. investigated the molecular profiling of 31 patients with lung cancer with comorbid IP, including 93.5% of smokers, and revealed *TP53*, *MET*, *BRAF*, and *KRAS* mutations in 64.5%, 12.9%, 9.7%, and 3.2%, respectively [64]. Masai et al. reported that in the surgically resected lung adenocarcinoma population, patients with the UIP pattern demonstrated lower *EGFR* mutation rates and higher *KRAS* mutation rates compared with those without the UIP pattern [65]. Honda et al. revealed *KRAS* mutations in 20.4% of patients with lung adenocarcinoma with a UIP pattern [42]. Additionally, Fujimoto et al. [41] and Ikeda et al. [66] reported *KRAS* mutations in 25% of NSCLC cases with comorbid IP. Table 1 summarizes the prevalence of driver gene mutations. Reports indicated bronchial metaplasia as a precancerous condition for IP-related lung adenocarcinoma [67,68], and another study detected a significant accumulation of *KRAS* mutations (G12V, G12C, and G12A) in the metaplastic epithelium [62]. These findings indicated a close relationship between lung adenocarcinoma and comorbid IP and *KRAS* mutations.

### 4.2. Pneumonitis Related to KRAS^G12C^ Inhibitors

Notably, the pivotal studies of sotorasib and adagrasib did not exclude patients with comorbid IP from the trials. However, the incidence of pneumonitis was relatively low. Regarding sotorasib, the phase II portion of CodeBreaK100 revealed pneumonitis as a treatment-related adverse event that occurred in two (1.6%) cases, one case each of grade 3 and grade 4 [57], and in the phase III trial of CodeBreaK200 [60], pneumonitis was observed in 1.8% of cases in the sotorasib group, compared with 2.0% in the docetaxel group. Concerning adagrasib, a phase II cohort of KRYSTAL-1 found any-grade pneumonitis in eight (6.9%) cases; three cases were grade ≥ 3 [61]. These data are very valuable for NSCLC with comorbid IP and indicate that *KRAS^G12C^* inhibitors may be tolerable in NSCLC with comorbid IP with a low acute exacerbation incidence.

### 4.3. Case Presentation

A 70-year-old woman with no history of smoking was referred to our hospital for the evaluation of a dry cough and an abnormal shadow on her chest X-ray. Chest CT revealed a 7-cm-diameter mass in the right lower lobe and two nodules in the right middle lobe, along with comorbid IP with the UIP pattern.

She underwent a transbronchial forceps biopsy of the primary lesion in the right lower lobe and was then diagnosed with lung adenocarcinoma. He underwent lobectomy of the right middle and lower lobes. Pathologically, R0 resection was achieved with a histopathologically clear margin, and she was diagnosed with invasive mucinous adenocarcinoma (non-treminal respiratory unit type, TTF-1 negative, HNF4a positive, p40 negative) at pathological stage IIIA (T4N0M0) with *KRAS^G12C^* mutation and PD-L1 of <1%. She received no adjuvant chemotherapy because of concerns about an acute exacerbation of IP.

Nine months after surgery, chest CT revealed multiple lung metastases, indicating a recurrence of invasive mucinous adenocarcinoma with comorbid IP. Carboplatin (area under the curve: 6, on day 1) and nab-paclitaxel (100 mg/m^2^, on days 1, 8, and 15) were initiated as a first-line regimen. However, she was evaluated as having progressive disease (PD) after two cycles. Subsequently, S-1 was initiated as a second-line regimen and continued for 13 months, achieving stable disease as the best overall response. However, she was diagnosed with PD due to increased bilateral lung metastases (Figure 2A−C). The patient began taking sotorasib as a third-line therapy at a daily dose of 960 mg. The best response was PR, and tumor shrinkage was maintained at 6 months (Figure 2D−F). Adverse events were grade 1 nausea and diarrhea, with no pneumonitis or acute exacerbation of pre-existing IP. Four months after the initiation of sotorasib, bilateral lung metastases were found to have reduced and were evaluated as PR, with the efficacy still continuing until the last outpatient visit, for 10 months since sotorasib started.

## 5. Conclusions and Future Perspectives

Treatment options for NSCLC with comorbid IP have been limited because of the risk of pharmacotherapy-induced acute exacerbations and the lack of sufficient evidence of efficacy. Platinum combination therapies, such as carboplatin plus nab-paclitaxel, are commonly used as first-line treatments for NSCLC with comorbid IP. However, there is currently no universally agreed standard of care for second-line and subsequent treatment due to a lack of sufficient evidence. IP is a poor prognostic comorbidity in patients with NSCLC. This is partly because typical driver gene mutations, such as *EGFR* mutations and *ALK* rearrangements, for which efficacy has been established in NSCLC with driver gene mutations, are rarely detected in the population of NSCLC with comorbid IP, and these mutations are more frequently found in nonsmokers. Even if one of these mutations is present, molecular-targeted therapies (especially *EGFR*-TKI) were hesitated in patients with comorbid IP because they may be at high risk for pneumonitis. Considering these backgrounds, exploration of driver gene mutations has frequently not been conducted in NSCLC patients with comorbid IP. Conversely, the presence of *KRAS* mutations has been reported to be potentially higher in NSCLC with comorbid IP than in NSCLC without IP because both *KRAS* mutations and IP are closely related to smoking. Hence, given the limited evidence for pharmacotherapy as described above, there is significant concern that valuable therapeutic opportunities with *KRAS^G12C^* inhibitors may have been overlooked for NSCLC patients with comorbid IP. As would be expected from the case presentation above, which demonstrates the safe use of *KRAS^G12C^* inhibitors for NSCLC with comorbid IP, it is hoped that further evidence regarding the use of *KRAS^G12C^* inhibitors for NSCLC with comorbid IP will continue to accumulate in the near future.

## Figures and Tables

**Figure 1 cancers-16-01327-f001:**
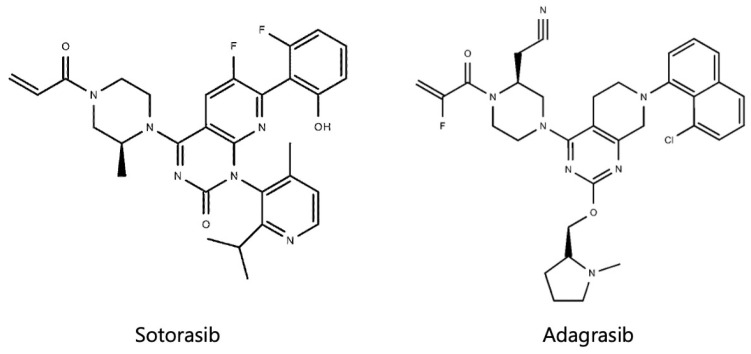
Structural formulas of sotorasib and adagrasib.

**Figure 2 cancers-16-01327-f002:**
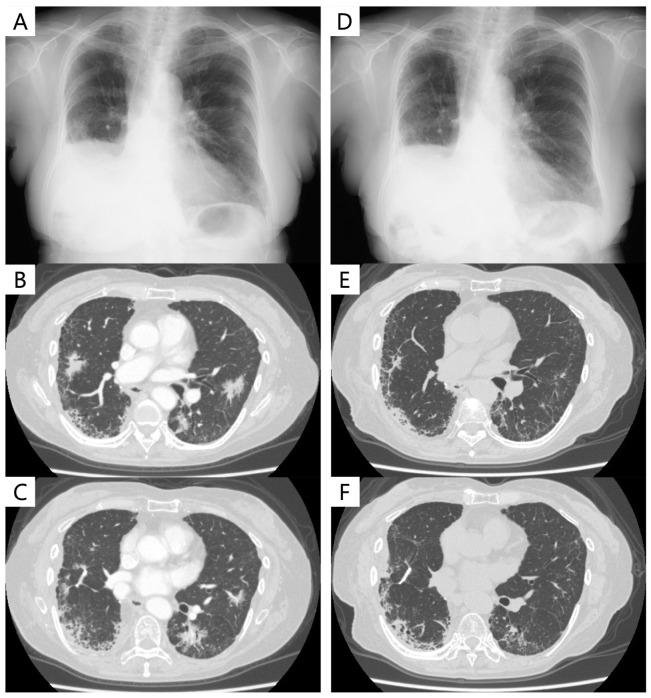
Chest X-ray and chest computer tomography (CT) of the presented case: (**A**) chest X-ray at the initiation of sotorasib; (**B**,**C**) chest CT at the initiation of sotorasib; (**D**) chest X-ray 6 months after sotorasib; (**E**,**F**) chest CT 6 months after sotorasib.

**Table 1 cancers-16-01327-t001:** Driver gene mutations detected in lung cancer patients with comorbid interstitial pneumonia.

Study	*EGFR*	*ALK*	*KRAS*	*BRAF*	Other
Fujimoto et al. [41]	1/31 (3.2%)	0/28 (0%)	7/28 (25.0%)	−	−
Honda et al. [42]	1/54 (1.9%)	0/54 (0%)	11/54 (20.4%)	2/54 (3.7%)	−
Primiani et al. [43]	9/65 (14.1%)	−	24/65 (37.5%)	−	−
Hwang et al. [44]	2/35 (5.7%)	0/35 (0%)	0/35 (0%)	6/35 (17.1%)	RET 2.7%
Guyard et al. [64]	1/31 (3.2%)	−	1/31 (3.2%)	3/31 (9.7%)	MET 12.9%
Ikeda et al. [66]	−	−	3/12 (25.0%)	−	−

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
