# Peer review of "KRASG12C Inhibitor as a Treatment Option for Non-Small-Cell Lung Cancer with Comorbid Interstitial Pneumonia"

_cancers, 2024, doi:10.3390/cancers16071327_

Round 1
Reviewer 1 Report
Comments and Suggestions for Authors
This is a quite long paper. Is it possible to reduce its first part, eventually summarizing data with a table? The final outcome of the case is not well reported.
As a case report, histology could be added.
Author Response
Response to the Reviewer #1’s comment
COMMENT: This is a quite long paper. Is it possible to reduce its first part, eventually summarizing data with a table? The final outcome of the case is not well reported.
As a case report, histology could be added.
RESPONSE: We appreciate the time and effort you invested in assisting us. Thank you for addressing this important point. We have revised the introductory section of our manuscript, condensing it to provide a more concise summary. Additionally, we have included the latest outcome and added more detailed histology regarding the presented case.
(Lines 45–61)
Non-small-cell lung cancer (NSCLC) with comorbid interstitial pneumonia (IP) presents limited treatment options and a poor prognosis due to the difficulties of treating NSCLC itself and the increased risk of acute exacerbation of IP with NSCLC treatment. Reportedly, 5%–10% of NSCLC patients have IP as a comorbidity at the time of diagnosis of NSCLC [1–3]. Conversely, the prevalence of lung cancer in idiopathic pulmonary fibrosis (IPF) ranges from 2.7%–48% [4], which is significantly higher than that in the general population [5]. This is due to the fact that the presence of IP, especially pulmonary fibrosis, has been shown to be associated with lung carcinogenesis [6]. The cumulative incidence rates of lung cancer in patients with IPF were 3.3%, 15.4%, and 54.7% at 1, 5, and 10 years, respectively. Additionally, the coexistence of emphysema may increase the risk of lung cancer development in patients with IPF [7,8]. Unlike the general population, squamous cell carcinoma is the most frequent type of lung cancer in patients with IPF [4,7]. Lung cancer is a poor prognostic factor in IPFwith 11% of all deaths attributed to it in Japan [9]. Furthermore, Kato et al. revealed that the 1-, 3-, and 5-year all-cause mortality rates in patients with IPF after a lung cancer diagnosis were 53.5%, 78.6%, and 92.9%, respectively [7]. Both IPF itself and the development of lung cancer demonstrated a significant effect on prognosis.
(Lines 62–79)
Acute exacerbation is an abrupt worsening of IP, sometimes clearly triggered by factors, such as infection or surgery, leading to high short-term mortality [10]. The criteria for diagnosing acute exacerbation in IPF proposed by an international working group report [11] are as follows: (1) previous or concurrent diagnosis of IPF, (2) acute worsening or development of dyspnea, typically within a 1-month duration, (3) computed tomog-raphy (CT) findings revealing new bilateral ground-glass opacity and/or consolidation superimposed on a background reticular shadow or honeycomb lung, and (4) deterio-ration that is not fully explained by cardiac failure or fluid overload. The annual incidence of acute exacerbation in IPF ranges from 8.5% to 14.2% [12–14], preceding up to 46% of deaths in IPF [9,11]. The median survival post-exacerbation is approximately 3–4 months, with high in-hospital mortality, frequently exceeding 50% [11]. Acute exacerbations in IP other than IPF have a poor prognosis, but are still better than IPF. Miyashita et al. [15] revealed that the 90-day mortality after acute exacerbation was 57% in IPF, 29% in nonIPF idiopathic IPs, and 33% in secondary IP. Patients with IP other than IPF demonstrated a significantly better survival rate than those with IPF after acute exacerbation (P < 0.001). Among those who survived, the proportion of patients needing long-term oxygen therapy post-exacerbation was 63% for IPF, 35% for non-IPF idiopathic IPs, and 46% for secondary IP.
(Lines 126–135)
The prevalence of ILA detected by CT has ranged from 8% to 22.8% in cohorts of smokers [31,32], and 13.2% of patients had ILA at the time of NSCLC diagnosis [33]. In a study involving patients with stage IV NSCLC, a high ILA score, which was visually evaluated, was associated with shorter survival [34]. Nakanishi et al. examined pretreatment chest CT scans for ILAs in 83 patients treated with anti-programmed death-1 (PD-1) antibodies such as nivolumab or pembrolizumab. They revealed a high incidence of immuno-therapy-associated pneumonitis at 17% (n = 14), with pre-existing ILAs correlating with a sixfold increase in the risk of drug-associated pneumonitis, especially those displaying a predominant ground-glass pattern of pneumonitis [35].
(Lines 351–354)
Pathologically, R0 resection was achieved with a histopathologically clear margin, and she was diagnosed with invasive mucinous adenocarcinoma (non-treminal respiratory unit type, TTF-1 negative, HNF4a positive, p40 negative) at pathological stage IIIA (T4N0M0) with KRASG12C mutation and PD-L1 of <1%.
(Lines 366–378)
Four months after the initiation of sotorasib, bilateral lung metastases found to be have reduced and were evaluated as PR, with the efficacy still continuing until the last out-patient visit, for 10 months since sotorasib started.
Reviewer 2 Report
Comments and Suggestions for Authors
The manuscript by Kazushi Fujimoto and co-authors presents a review on the possibilities of chemotherapy for patients with non-small-cell lung cancer (NSCLC) and comorbid interstitial pneumonia (IP). The authors, in particular, analyzed the literature data on driver gene mutations in NSCLC/IP that are targeted by different chemotherapeutic agents. The data showed that KRAS and BRAF driver gene mutations were often observed in NSCLC patients with comorbid IP. This observation opens new treatment opportunities. The application of several KRASG12C inhibitors in treating NSCLC with comorbid IP was described, including the case study of an elderly patient. The authors show safety and perspectives of KRASG12C inhibitors for NSCLC/IP. In my opinion, the manuscript presents a critical analysis of the literature data in the field of cancer treatment and deserves publication. The literature cited is up-to-date, representative, and relevant.
Specific comments:
Line 363: Obviously, the phrase "He underwent lobectomy" should be replaced with "She underwent lobectomy".
Page 7, Section Conflict of interests: It is unclear (at least for me), why there can be a conflict concerning fundings obtained "outside of the submitted work".
Summarizing, I recommend acceptance of the manuscript for publication after minor revision.
Author Response
Response to the Reviewer #2’s comment
COMMENT: The manuscript by Kazushi Fujimoto and co-authors presents a review on the possibilities of chemotherapy for patients with non-small-cell lung cancer (NSCLC) and comorbid interstitial pneumonia (IP). The authors, in particular, analyzed the literature data on driver gene mutations in NSCLC/IP that are targeted by different chemotherapeutic agents. The data showed that KRAS and BRAF driver gene mutations were often observed in NSCLC patients with comorbid IP. This observation opens new treatment opportunities. The application of several KRASG12C inhibitors in treating NSCLC with comorbid IP was described, including the case study of an elderly patient. The authors show safety and perspectives of KRASG12C inhibitors for NSCLC/IP. In my opinion, the manuscript presents a critical analysis of the literature data in the field of cancer treatment and deserves publication. The literature cited is up-to-date, representative, and relevant.
RESPONSE: We appreciate you taking the time and effort to offer us your comments and insights related to the paper. In the following sections, you will find our responses to each of your points and suggestions.
Specific comments:
Line 363: Obviously, the phrase "He underwent lobectomy" should be replaced with "She underwent lobectomy".
RESPONSE: Thank you for pointing out the grammatical error. We have corrected it as you mentioned.
Page 7, Section Conflict of interests: It is unclear (at least for me), why there can be a conflict concerning fundings obtained "outside of the submitted work".
Summarizing, I recommend acceptance of the manuscript for publication after minor revision.
RESPONSE: Thank you for pointing out this important point. In the instructions for authors provided by Cancers, the section on conflict of interest emphasizes the need for authors to disclose all relationships or interests that could potentially influence or bias their work. Additionally, the MDPI disclosure form highlights the importance of transparency in disclosing any relationships or activities, even those outside of the submitted work, as perceptions of conflict can impact trust in science.
Therefore, we considered conflicts of interest outside of the submitted work to be relevant for ensuring transparency and maintaining trust in the scientific process. Thank you for your understanding and cooperation.
Reviewer 3 Report
Comments and Suggestions for Authors
The article titled “KRASG12C Inhibitor as a Treatment Option for Non-small-cell 2 Lung Cancer with Comorbid Interstitial Pneumonia” has been evaluated. This work requires significant revision in order to attract the attention of a general audience. While the authors have accomplished a great deal in terms of presenting the most recent articles on the topic, the presentation of their findings needs to be improved in order to make it more engaging for a broader audience. To achieve this goal, the authors need to focus on presenting their figures and data in a way that can hold the interest of the general readership of this journal. This means moving beyond the technical details and finding ways to make the findings more accessible and engaging for those who may not have the same level of expertise in the area. One approach that the authors could take is to consider how readers can get the most out of the figures without having to read through all of the text. They could also find ways to maintain the readers' interest after they have gone through most of the figures. This would require the authors to present their findings in a way that is both informative and visually stimulating. Another approach that the authors might consider is to include several graphic figures of biological studies. Obtaining copyright permission for these figures should be easy, and this would help attract the interest of the general audience of the journal. Additionally, the authors should also include some chemical structures of KRASG12C inhibitors in order to provide readers with a better understanding of the scientific principles behind their research. Finally, the authors should provide a brief description of the current treatment strategies for this type of cancer so that readers can better understand the significance of their findings. By taking these steps, the authors can significantly improve the presentation of their work and make it more accessible and engaging for a wider audience.
Comments on the Quality of English LanguageModerate editing of English language required
Author Response
Response to the Reviewer #3’s comment
The article titled “KRASG12C Inhibitor as a Treatment Option for Non-small-cell Lung Cancer with Comorbid Interstitial Pneumonia” has been evaluated. This work requires significant revision in order to attract the attention of a general audience. While the authors have accomplished a great deal in terms of presenting the most recent articles on the topic, the presentation of their findings needs to be improved in order to make it more engaging for a broader audience. To achieve this goal, the authors need to focus on presenting their figures and data in a way that can hold the interest of the general readership of this journal. This means moving beyond the technical details and finding ways to make the findings more accessible and engaging for those who may not have the same level of expertise in the area. One approach that the authors could take is to consider how readers can get the most out of the figures without having to read through all of the text. They could also find ways to maintain the readers' interest after they have gone through most of the figures. This would require the authors to present their findings in a way that is both informative and visually stimulating. Another approach that the authors might consider is to include several graphic figures of biological studies. Obtaining copyright permission for these figures should be easy, and this would help attract the interest of the general audience of the journal. Additionally, the authors should also include some chemical structures of KRASG12C inhibitors in order to provide readers with a better understanding of the scientific principles behind their research. Finally, the authors should provide a brief description of the current treatment strategies for this type of cancer so that readers can better understand the significance of their findings. By taking these steps, the authors can significantly improve the presentation of their work and make it more accessible and engaging for a wider audience.
Moderate editing of English language required.
RESPONSE: We appreciate you taking the time and effort to offer us your comments and insights related to the paper and are pleased to inform you of the revisions made in response to your suggestions.
Firstly, we have included additional chemical structures of KRASG12C inhibitors (Figure 1) in the manuscript to improve the comprehension of the scientific principles behind our research. These additions not only enhance the clarity of our findings but also aim to engage a broader readership on a deeper level.
Secondly, we have addressed the suggestion to include a brief description of current treatment strategies for NSCLC with comorbid IP in the conclusion section. By doing so, we aim to provide readers with a better context for understanding the significance of our findings in the broader clinical landscape.
Regarding the English language, we want to assure you that our manuscript has undergone meticulous language editing to ensure clarity and coherence. We have worked with professional language editors to refine the text and eliminate any grammatical errors or language inconsistencies.
We are confident that these revisions have significantly improved the presentation of our work, making it more accessible and engaging for a wider audience. We believe that the incorporation of additional structural formulas and the inclusion of pertinent information in the conclusion will enhance the overall quality of the manuscript.
Once again, we appreciate the opportunity to revise our article based on your insightful feedback. We hope that these modifications meet your expectations and look forward to the possibility of our article being reconsidered for publication in your esteemed journal.
Thank you for your time and consideration.
(Lines 242–244)
Sotorasib and adagrasib (Fig. 1) were approved as second-line treatment options for advanced or recurrent NSCLC with KRASG12C mutations.
(Lines 377–380)
Platinum combination therapies, such as carboplatin plus nab-paclitaxel, are commonly used as first-line treatment for NSCLC with comorbid IP. However, there is currently no universally agreed standard of care for second-line and subsequent treatment due to a lack of sufficient evidence.
(Lines 391–393)
Hence, given the limited evidence for pharmacotherapy as described above, there is significant concern that valuable therapeutic opportunities with KRASG12C inhibitors may have been overlooked for NSCLC patients with comorbid IP.
Round 2
Reviewer 1 Report
Comments and Suggestions for Authors
I accept the corrected test.